# Urine Di-(2-ethylhexyl) Phthalate Metabolites Are Independently Related to Body Fluid Status in Adults: Results from a U.S. Nationally Representative Survey

**DOI:** 10.3390/ijerph19126964

**Published:** 2022-06-07

**Authors:** Wei-Jie Wang, Chia-Sung Wang, Chi-Kang Wang, An-Ming Yang, Chien-Yu Lin

**Affiliations:** 1Division of Nephrology, Department of Internal Medicine, Taoyuan General Hospital, Ministry of Health and Welfare, Taoyuan 330, Taiwan; mrwwj@tygh.gov.tw; 2Department of Biomedical Engineering, Chung Yuan Christian University, Taoyuan 300, Taiwan; 3Department of Internal Medicine, En Chu Kong Hospital, New Taipei City 237, Taiwan; 01143@km.eck.org.tw (C.-S.W.); 01189@km.eck.org.tw (A.-M.Y.); 4Department of Environmental Engineering and Health, Yuanpei University of Medical Technology, Hsinchu 300, Taiwan; ckwang@mail.ypu.edu.tw; 5Department of Nursing, Yuanpei University of Medical Technology, Hsinchu 300, Taiwan; 6School of Medicine, Fu Jen Catholic University, New Taipei City 242, Taiwan

**Keywords:** di-(2-ethylhexyl) phthalate, bioelectrical impedance analysis, extracellular fluid, National Health and Nutrition Examination Survey, obesity

## Abstract

Purpose: Di-(2-ethylhexyl) phthalate (DEHP) has been utilized in many daily products for decades. Previous studies have reported that DEHP exposure could induce renin–angiotensin–aldosterone system activation and increase epithelial sodium channel (ENaC) activity, which contributes to extracellular fluid (ECF) volume expansion. However, there is also no previous study to evaluate the association between DEHP exposure and body fluid status. Methods: We selected 1678 subjects (aged ≥18 years) from a National Health and Nutrition Examination Survey (NHANES) in 2003–2004 to determine the relationship between urine DEHP metabolites and body composition (body measures, bioelectrical impedance analysis (BIA)). Results: After weighing the sampling strategy in multiple linear regression analysis, we report that higher levels of DEHP metabolites are correlated with increases in body measures (body weight, body mass index (BMI), waist circumference), BIA parameters (estimated fat mass, percent body fat, ECF, and ECF/intracellular fluid (ICF) ratio) in multiple linear regression analysis. The relationship between DEHP metabolites and the ECF/ICF ratio was more evident in subjects of younger age (20–39 years old), women, non-Hispanic white ethnicity, and subjects who were not active smokers. Conclusion: In addition to being positively correlated with body measures and body fat, we found that urine DEHP metabolites were positively correlated with ECF and the ECF/ICF ratio in the US general adult population. The finding implies that DEHP exposures might increase ECF volume and the ECF/ICF ratio, which may have adverse health outcomes on the cardiovascular system. Further research is needed to clarify the causal relationship.

## 1. Introduction

Phthalates, known as endocrine-disrupting chemicals (EDCs), are the most common chemicals widely used in many products. They are added to increase the flexibility of the plastic due to their hydrophobicity and non-covalent bonding of the polymer [1,2]. After exposure, phthalates undergo a series of phase I hydrolysis and phase II conjugation reactions, and their metabolites are excreted in feces and urine [3]. People are widely exposed to phthalates because their metabolites are commonly detected in urine samples around the world [4]. Although the half-life of phthalates is only approximately 12 h [5], due to continuous exposure, their effects are similar to those of persistent and bioaccumulating compounds [6]. Among the compound esters of phthalates, di-(2-ethylhexyl) phthalate (DEHP) is quantitatively one of the most important members of the phthalate ester group and has been listed in Annex XIV of REACH of the European Union since 2015 [7]. Recently, several epidemiological studies have linked DEHP exposure to several cardiovascular disease risk factors, such as diabetes mellitus [8] and higher systolic blood pressure [9].

In healthy adults, extracellular fluid (ECF) content comprises ~33–40% of total body water [10] and is regulated by the activity of the renin–angiotensin–aldosterone system (RAAS), autonomic nervous system, hormones, and natriuretic peptides [11]. The increased ECF might cause stress in the cardiovascular system, which eventually contributes to hypertension [12] and has been reported as an independent predictor of cardiovascular morbidity in patients with chronic renal failure [13]. The ECF/intracellular fluid (ICF) volume ratio also shows a significant association with chronic disease [14,15,16,17]. Recently, many environmental chemicals were reported to influence blood pressure and ECF [18]. The mechanism was related to the activation of RAAS by inhibiting 11β hydroxysteroid dehydrogenase and type 2 (11β-HSD-2) enzyme activity [18]. The activation of RAAS contributes to sodium retention and ECF expansion [19]. Previous animal and human studies have reported that DEHP exposure is associated with the activation of the RAAS by inhibiting 11β-HSD-2 [20,21]. Moreover, DEHP exposure might increase the risk of obesity [22], while adipocytes are an important source of extra-adrenal aldosterone [23]. In addition, DEHP has been reported to increase epithelial sodium channel (ENaC) activity in renal cortical collecting duct cells [21] and have an affinity with the receptors of peroxisome proliferator-activated receptor (PPAR) γ [24], which can expand the ECF volume in turn [25,26]. However, no previous epidemiological study has investigated the relationship between DEHP exposure and fluid status.

Bioimpedance analysis (BIA) is a tool for measuring the electrical impedance of human tissues and can be used to assess the proportion of lean mass, fat, and body fluids [27]. Unlike fluid volume, the association between DEHP exposure and other body measures has been extensively studied. When using body mass index (BMI), weight, and waist circumference as parameters, some results showed that higher levels of urine DEHP metabolites showed a positive correlation [28,29,30], while others showed no correlation [31] or an inverse association [32]. The relationship between DEHP exposure and body fat percentage has been most investigated in children, and the results are inconsistent [33,34,35,36]. In adults, however, there are only a few studies studying the association between DEHP exposure and lean mass/body fat percentage, and the results are also inconsistent [37,38].

Since the relationship between DEHP exposure and fluid status is unknown, we included subjects enrolled in the National Health and Nutrition Examination Survey (NHANES). In the NHANES database, BIA measurements were available between 1999 and 2004. However, detailed urine DEHP metabolites were only available in NHANES 2003–2004. We hypothesized that urine DEHP metabolites may have a positive correlation with body measures, lean mass, fat mass, and body fluids in the NHANES 2003–2004 database, which has detailed information on urine DEHP metabolites, body measurements, BIA, and various covariates.

## 2. Materials and Methods

### 2.1. Study Design and Population

The data for this study were obtained from NHANES 2003–2004. NHANES is a study of a representative sample of the U.S. population that collects information about family health and nutrition. The survey data are released every two years. Detailed contents of the NHANES 2003–2004 are available on the NHANES website [39]. Our analysis included 1678 participants over the age of 18 without missing data on basic demographics and BMI and those who had undergone urine DEHP metabolite testing. All methods were carried out in accordance with relevant guidelines and regulations.

### 2.2. Urine DEHP Metabolites

In NHANES 2003–2004, urine DEHP metabolites, including mono (2-ethylhexyl) phthalate (MEHP), mono(2-ethyl-5-oxohexyl) phthalate (MEOHP), mono(2-ethyl-5-hydroxyhexyl) phthalate (MEHHP), and mono(2-ethyl-5-carboxypentyl) phthalate (MECPP), were tested using high-performance liquid chromatography–electrospray ionization–tandem mass spectrometry. For concentrations below detection limits, a value was assigned by NHANES. We used this value in our analyses. The detailed method is available in the Appendix A.

### 2.3. Body Measures

The body measurement assessments performed on survey participants varied according to the participants’ ages. Weight, BMI, triceps, and subscapular skinfold were measured in NHANES 2003–2004. The detailed method is available in the Appendix A.

### 2.4. Bioelectrical Impedance Analysis (BIA)

In NHANES 2003–2004, this examination was conducted in an eligible survey with participants 8–49 years of age. The BIA data were collected with a HYDRA ECF/ICF Bio-Impedance Spectrum Analyzer (Model 4200, Xitron Technologies, Inc., San Diego, CA, USA). Detailed information is available in the Appendix A.

### 2.5. Covariates

We considered age, gender, race/ethnicity, education level, household income, smoking status, caffeine intake, total energy intake, total protein intake, total sugar intake, total carbohydrate intake, total saturated fatty acid intake, and metabolic equivalent intensity level of activity to be potential confounders in this study. Detailed information is available in the Appendix A.

### 2.6. Statistics

DEHP metabolite concentrations were corrected for urine creatinine and expressed as the geometric mean (geometric standard error) in different subpopulations. These variables were tested by Student’s 2-tailed *t*-test and one-way analysis of variance. As DEHP is metabolized primarily into MEHP, MEOHP, MEHHP, and MECPP, we divided the level of each metabolite by its molar mass and then added the concentrations of each metabolite to calculate Σ DEHP [40]. Due to the obvious deviation from the normal distribution, the DEHP metabolites were transformed by a natural logarithm. We constructed an extended model approach with body measures/bioelectrical impedance analysis as the dependent variable and individual ln-DEHP metabolites as a predictor. Model 1 adjusted for age, gender, race and ethnicity, education level, household income, smoking status, and metabolic equivalent intensity level for activity. Model 2 adjusted for Model 1 plus caffeine intake, total energy intake, total protein intake, total sugar intake, total carbohydrate intake, and total saturated fatty acid intake.

We used sample weights for analysis to understand the impact of weights. The calculation of sampling weight follows the analysis guidelines of the National Center for Health Statistics and appropriately considers the complex survey design adopted in NHANES 2003–2004 [41]. All analyses were calculated using SPSS Version 20 (SPSS Inc. Chicago, IL, USA). *p* < 0.05 was considered significant.

## 3. Results

MEHP, MEOHP, MEHHP, and MECPP were detectable in 69.9%, 99.1%, 99.7%, and 100% of study subjects, respectively. Table 1 shows the basic demographics of the study subjects. The study participants were composed of 798 men and 880 women. Subjects aged between 18 and 39 years old, women, and those with higher education levels were associated with higher DEHP metabolite concentrations and Σ DEHP. In addition to MEHP, subjects with higher income were associated with higher levels of other DEHP metabolites and Σ DEHP, while these metabolite levels were different between races. Participants with a BMI between 25 and 30 had a lower concentration of DEHP metabolites and Σ DEHP, while active smokers had a lower concentration of MEOHP, MECPP, and Σ DEHP. Those who had a body fat percentage ≥35% had higher levels of MEOHP, MECPP, and Σ DEHP.

The means and S.E. of body fluid status between the basic demographic of the sample subjects are shown in Table 2. Male gender, higher household income, higher BMI, lower body fat percentage, active smoking, and higher alcohol consumption were associated with higher ECF and ICF volumes. A higher ECF volume was also found in subjects who were non-Hispanic white. Subjects aged between 40 and 59 years old, female gender, higher education level, lower BMI, higher body fat percentage, and lower alcohol consumption were associated with a higher ECF/ICF ratio, while a lower ECF/ICF ratio was found in other Hispanics.

After weighing for sampling strategy in multiple linear analysis, the correlations between DEHP metabolite levels and various body component measures are shown in Table 3. Levels of MEOHP, MEHHP, MECPP, and Σ DEHP were positively associated with weight, BMI, and waist circumference. The correlations between DEHP metabolite levels and parameters of bioelectrical impedance analysis are shown in Table 4. The levels of MEOHP, MEHHP, MECPP, and Σ DEHP were positively correlated with estimated fat mass, estimated percent body fat, ECF, and the ECF/ICF ratio.

Linear regression coefficients between Σ DEHP, BMI, percent fat mass, and ECF/ICF ratio in different subpopulations are demonstrated in Table 5. The correlations between Σ DEHP levels and BMI were more evident in subjects with older age (≥40 years old) and nonactive smokers. The relationship between Σ DEHP and estimated percent body fat and the ECF/ICF ratio was more evident in subjects of younger age (20–39 years old), women, and non-Hispanic white ethnicity. In addition, the associations between Σ DEHP and the ECF/ICF ratio were also evident in subjects who were not active smokers.

## 4. Discussion

In the current study, we reported that the levels of DEHP metabolites were positively correlated with parameters of body measures (weight, BMI, and waist circumference) and parameters of BIA (fat mass, estimated percent body fat, ECF, and ECF/ICF ratio). Although the relationship between DEHP exposure and obesity parameters has been extensively studied, this is the first report identifying a positive association between urine DEHP metabolites and ECF and the ECF/ICF ratio in a nationally representative survey of U.S. adults. The representativeness of the study population is the main advantage of this study, and we were able to control many covariates in the comprehensive NHANES database.

BMI and waist circumference are generally used as screening tools for being overweight or obese with high specificity [22]. However, these two parameters are less sensitive in identifying adiposity because they cannot distinguish half of the people with high body fat [42]. Some experts suggest the value of assessing body-fat percentage, which is more direct for measuring adiposity [42]. In the current study, subjects with higher BMI and lower body fat percentage were associated with higher ECF and ICF volumes and lower ECF/ICF ratio. This discrepancy may explain by adipocytes produce 30% of the total blood angiotensinogen [43] and are also an important source of extra-adrenal aldosterone [23]. It is possible that those with higher body fat produce more angiotensinogen and aldosterone to increase ECF more and result in a higher ECF/ICF ratio. Subjects with higher BMI do not necessarily have higher body fat, as the higher BMI may due to more fat-free mass (e.g., muscle). Water is the main component of muscle, accounting for about 76% of its mass [44]. Higher muscle mass increases the proportion of ICF than ECF in body composition, which results in a lower ECF/ICF ratio.

Recently, it has been discovered that the effects of environmental toxins, especially those defined as EDCs, can promote obesity and are called “obesogens” [28,45]. In animal studies, female C3H/N mice exposed to DEHP doses similar to environmental exposures showed increased food intake, body weight, and visceral fat deposits [46]. In male C3H/He mice, chronic exposure to DEHP may induce obesity through changes in energy homeostasis. The synergistic effect of hypothyroidism and hypothalamic leptin resistance is a possible mechanism [47]. Several recent epidemiological studies have surveyed the association between DEHP exposure and obesity in adults. All the previous studies used BMI/waist/weight gain as markers of obesity. In cross-sectional studies, one study observed that higher MEHP levels were associated with increased obesity prevalence in NHANES 1999–2004 [29]. Another study using data from the 2007–2010 NHANES reported that higher levels of MECPP, MEHHP, and Σ DEHP were associated with an increased prevalence of obesity [28]. However, other studies found that MEHP levels have inverse associations with obesity in both 1999–2002 NHANES data [32] and the Nurses’ Health Study (NHS) and NHS2 cohorts [31]. By using a physiologically based pharmacokinetic model, a study reported that the relationship between BMI and DEHP may be explained by higher energy intake and consequent higher DEHP exposure [48]. In prospective analyses, MEOHP levels had positive trends with weight gain between baseline and year 3. However, no statistically significant association was observed after 6 years [30]. In the current study, we found that higher levels of DEHP metabolites were correlated with higher body measure parameters (weight, BMI, and waist circumference). Differences between studies may be due to a variety of factors, such as race, measurement method, lifestyle, and food.

Previous studies on DEHP exposure and fat mass were limited to children. Some studies reported that there was no correlation between DEHP exposure and adiposity [33,35,36]. Another report showed that the associations of DEHP exposure with body fat depended on the timing of exposure [34]. In adults, there were few studies studying the association between DEHP exposure and body fat percentage. One study found no correlation between urine MEHP levels and adiposity measures in elderly women [38]. Another study enrolled participants from NHANES 1999–2006, and the authors reported that there was no association between MEHP and body fat percentage but a negative correlation with lean mass among women [37]. The difference between our study and the above two studies is that we used many metabolites of DEHP as biological indicators instead of MEHP alone. In this study, we found that the levels of MEOHP, MEHHP, MECPP, and Σ DEHP were positively correlated with estimated fat mass and estimated percent body fat. The results provided evidence that DEHP exposure might increase fat mass/body fat percentage in adults.

The fluid volume in humans depends on age, sex, and body size and is also associated with human health. ECF is controlled by sodium balance and total body sodium content. The osmolality of the ECF is regulated by water intake and vasopressin secretion, while the ECF volume is maintained by RAAS and some natriuretic factors, including atrial/brain natriuretic peptide [11]. In addition, growth and sex hormones have been found to contribute to ECF volume [25,49]. Environmental chemicals that have the capacity to inhibit 11β-HSD-2 would be predicted to elevate blood pressure and expand ECF [18]. In mice, DEHP exposure has been reported to elevate blood pressure through activation of the RAAS [20]. In premature infants, higher levels of postnatal DEHP exposure are correlated with increased blood pressure and hypertension. Statistical analysis showed that this relationship was mediated by the cortisol-to-cortisone ratio, suggesting that the mechanism may be because DEHP inhibits the activity of 11β-HSD-2 and then activates the mineralocorticoid receptor. In this study, increased expression of ENaC and phosphorylated Na^+^–Cl^−^ cotransporters were also found in hypertensive infants [21]. Moreover, we reported DEHP exposure is positively associated with fat mass, which might increase the production of angiotensinogen [10] and aldosterone [3]. It is possible that DEHP induces activation of RAAS and then contributes to sodium retention and ECF expansion [19]. In addition, the mono- and dicarboxylic acid metabolites of DEHP have affinities with the receptors of peroxisome proliferator-activated receptor (PPAR)α and PPARγ in vitro studies [24]. Activation of PPARγ results in enhanced sodium and water reabsorption by upregulation of sodium–hydrogen exchanger 3, the Na-K-2Cl cotransporter, water channels aquaporin-2, and ENaC [26]. Another possible mechanism is that DEHP has been reported to have estrogenic activity both in vitro and in vivo [50]. The high estrogen status will then increase ENaC activity via protein kinase C δ signaling in renal cortical collecting duct cells and expand the ECF volume [25]. In the current study, we provide the first evidence that the levels of DEHP metabolites are positively correlated with ECF and the ECF/ICF ratio. Increased ECF may cause stress on the cardiovascular system [13], which may be one of the possible mechanisms by which DEHP causes cardiovascular disease. However, this hypothesis still needs further study.

In the subgroup analysis, the correlation between Σ DEHP and the ECF/ICF ratio was more evident in women. Women are exposed to a higher concentration of phthalates than men because of the personal care products they use [51]. Furthermore, a report showed that there were sex differences in the regulation of RAAS. The adrenal response to exogenous angiotensin II was significantly higher in women. It is possible that the effect of DEHP-induced RAAS may be more evident in women than in men [52]. We also found that the association between Σ DEHP and the ECF/ICF ratio was more pronounced among subjects who were not current smokers. Studies show that body composition may be altered by smoking habits [53,54]. A possible explanation is that DEHP has a much weaker effect on body fluids than tobacco smoke. When considering the impact of DEHP on active smokers, the trend is too small to be statistically significant.

Our research has several limitations. First, causal inference is not suitable for cross-sectional research. Second, when DEHP is exposed, other undetected chemicals may be exposed at the same time, and the correlation found in the research is not caused by DEHP itself. Third, our study population was mainly composed of adults, so we cannot infer that this association also exists in children. Finally, the relatively small sample size, especially in subgroup analysis, is a limitation of the study. Our results need to be interpreted with caution.

## 5. Conclusions

Although previous reports have explored the association between DEHP exposure and obesity parameters, including BMI, waist circumference, and fat percentage in adults, we present the first report identifying a positive association of urine DEHP metabolites with body fluid status in a nationally representative survey of U.S. adults. If a causal relationship exists, DEHP exposure might increase ECF volume and the ECF/ICF ratio through its affinity for PPAR γ receptor [24], as well as increasing the production of extra-adrenal aldosterone [23], increasing ENaC activity [21], and inhibiting 11β-HSD-2 activity [20,21]. Our results provide important clues for future toxicological research. Since DEHP exposure has become a worldwide concern, further research is necessary to determine the long-term mechanism and effects of low-dose DEHP exposure on human health.

## Figures and Tables

**Table 1 ijerph-19-06964-t001:** Basic demographics of the sample subjects, including geometric means (geometric S.E.) of DEHP metabolites concentrations.

	Unweighted No. (%)	MEHP (μg/g Creatinine)	MEOHP (μg/g Creatinine)	MEHHP (μg/g Creatinine)	MECPP (μg/g Creatinine)	Σ DEHP (μmol/g Creatinine)
Overall	1678 (100)	2.13 (1.03)	12.18 (1.03)	18.23 (1.03)	29.44 (1.03)	0.21(1.03)
Age, year						
18–39	718 (42.8)	2.64 (1.05) ^‡^	13.63 (1.05) ^‡^	20.15 (1.05) ^†^	30.88 (1.04) ^†^	0.23 (1.04) ^†^
40–59	449 (26.8)	1.90 (1.06) ^‡^	10.67 (1.05) ^‡^	16.33 (1.05) ^†^	25.89 (1.05) ^†^	0.19 (1.05) ^†^
≥60	511 (30.4)	1.72 (1.05) ^‡^	11.70 (1.04) ^‡^	17.46 (1.04) ^†^	30.81 (1.04) ^†^	0.21 (1.04) ^†^
Gender						
Men	798 (47.6)	1.85 (1.05) ^‡^	10.74 (1.04) ^‡^	16.52 (1.04) ^‡^	26.08 (1.04) ^‡^	0.19 (1.04) ^‡^
Women	880 (52.4)	2.41 (1.04) ^‡^	13.66 (1.04) ^‡^	19.94 (1.04) ^‡^	32.86 (1.03) ^‡^	0.24 (1.03) ^‡^
Race						
Mexican American	365 (21.8)	1.96 (1.06)	10.36 (1.06) *	15.18 (1.06) *	26.30 (1.05) *	0.19 (1.05) *
Other Hispanic	48 (2.9)	2.63 (1.17)	12.92 (1.17) *	19.59 (1.18) *	32.47 (1.14) *	0.24 (1.15) *
Non-Hispanic White	842 (50.2)	2.09 (1.04)	12.91 (1.04) *	19.19 (1.04) *	32.00 (1.04) *	0.23 (1.04) *
Non-Hispanic Black	345 (20.6)	2.37 (1.07)	12.72 (1.06) *	19.64 (1.06) *	26.91 (1.06) *	0.21 (1.06) *
Others	78 (4.5)	2.02 (1.14)	11.09 (1.13) *	17.09 (1.13) *	28.41 (1.12) *	0.20 (1.12) *
Education levels						
≤High school	927 (55.3)	1.94 (1.04) ^‡^	10.85 (1.03) ^‡^	16.04 (1.04) ^‡^	26.54 (1.03) ^‡^	0.19 (1.03) ^‡^
>High school	750 (44.7)	2.37 (1.05) ^‡^	14.06 (1.04) ^‡^	21.35 (1.04) ^‡^	33.47 (1.04) ^‡^	0.25 (1.04) ^‡^
Annual household income						
<USD 25,000	556 (35.6)	2.02 (1.05)	10.98 (1.05) ^‡^	16.50 (1.05) ^‡^	27.52 (1.04) ^†^	0.20 (1.04) ^‡^
USD 25,000–55,000	503 (32.2)	2.00 (1.05)	11.54 (1.05) ^‡^	17.24 (1.05) ^‡^	27.89 (1.05) ^†^	0.20 (1.05) ^‡^
>USD 55,000	501 (32.2)	2.31 (1.06)	14.07 (1.05) ^‡^	21.06 (1.05) ^‡^	33.01 (1.05) ^†^	0.24 (1.05) ^‡^
BMI, kg/m^2^						
<25	577 (34.4)	2.31 (1.05) *	11.92 (1.05) ^‡^	17.77 (1.05) ^‡^	28.71 (1.05) *	0.21 (1.05) ^‡^
25–30	562 (33.5)	1.90 (1.05) *	10.93 (1.05) ^‡^	16.31 (1.05) ^‡^	27.29 (1.04) *	0.20 (1.04) ^‡^
≥30	539 (32.1)	2.19 (1.05) *	13.97 (1.05) ^‡^	21.06 (1.05) ^‡^	32.73 (1.04) *	0.24 (1.04) ^‡^
Body fat percentage (%)						
<25	227 (31.4)	2.22 (1.09)	10.70 (1.08) *	16.81 (1.09)	25.82 (1.08) *	0.19 (1.08) *
25–35	230 (31.9)	2.00 (1.09)	11.33 (1.08) *	17.11 (1.08)	25.87 (1.07) *	0.20 (1.07) *
≥35	265 (36.7)	2.53 (1.08)	14.19 (1.07) *	20.83 (1.08)	32.21 (1.07) *	0.24 (1.07) *
Smoking						
Nonexposed	353 (21.0)	2.16 (1.07)	12.10 (1.06) *	18.26 (1.06)	31.27 (1.05) ^‡^	0.22 (1.06) *
Expose to ETS	849 (50.6)	2.15 (1.04)	12.99 (1.04) *	19.04 (1.04)	31.15 (1.04) ^‡^	0.23 (1.04) *
Active smokers	476 (28.4)	2.07 (1.06)	10.93 (1.05) *	18.23 (1.05)	25.46 (1.05) ^‡^	0.19 (1.05) *
Alcohol consumption (drinks/year)						
<12	412 (29.5)	2.01 (1.06)	11.86 (1.05)	17.38 (1.05)	2.52 (1.05)	0.21 (1.05)
≥12	983 (70.5)	2.05 (1.04)	11.89 (1.04)	17.97 (1.04)	2.32 (1.03)	0.21 (1.03)

*, *p* < 0.05; ^†^, *p* < 0.01; ^‡^, *p* < 0.005 (tested with Student’s 2-tailed *t*-test or by one-way analysis of variance); Abbreviations: DEHP, di-(2-ethylhexyl) phthalate; ETS, environmental tobacco smoke; MEHP, mono(2-ethylhexyl) phthalate; MEHHP, mono(2-ethyl-5-hydroxyhexyl) phthalate; MEOHP, mono(2-ethyl-5-oxohexyl) phthalate; MECPP, mono(2-ethyl-5-carboxypentyl) phthalate. Σ DEHP was the sum of (MEHP/278) + (MEHHP/294) + (MEOHP/292) + (MECPP/308) and corrected for urine creatinine.

**Table 2 ijerph-19-06964-t002:** Basic demographics of the sample subjects, including means (S.E.) of body fluid status.

	Unweighted No. (%)	ECF (L)	ICF (L)	ECF/ICF Ratio
Overall	726 (100)	16.92 (0.13)	23.35 (0.24)	0.74 (0.00)
Age, year				
18–39	524 (42.8)	16.80 (0.16)	23.42 (0.29)	0.74 (0.00) ^‡^
40–50	202 (26.8)	17.24 (0.24)	23.16 (0.42)	0.76 (0.01) ^‡^
Gender				
Men	377 (47.6)	19.00 (0.15) ^‡^	27.61 (0.28) ^‡^	0.70 (0.00) ^‡^
Women	349 (52.4)	14.67 (0.15) ^‡^	18.74 (0.22) ^‡^	0.79 (0.00) ^‡^
Race				
Mexican American	172 (21.8)	15.95 (0.23) ^‡^	22.32 (0.43)	0.73 (0.01) *
Other Hispanic	27 (2.9)	15.99 (0.64) ^‡^	22.92 (1.28)	0.72 (0.02) *
Non-Hispanic White	319 (50.2)	17.45 (0.21) ^‡^	23.78 (0.38)	0.75 (0.01) *
Non-Hispanic Black	172 (20.6)	17.13 (0.27) ^‡^	23.78 (0.49)	0.74 (0.01) *
Others	36 (4.5)	16.59 (0.65) ^‡^	22.70 (1.22)	0.75 (0.01) *
Education levels				
≤High school	376 (55.3)	16.88 (0.18)	23.58 (0.33)	0.73 (0.00) ^‡^
>High school	350 (44.7)	16.97 (0.20)	23.10 (0.36)	0.75 (0.00) ^‡^
Annual household income				
<USD 25,000	203 (35.6)	16.40 (0.23) ^‡^	22.50 (0.41) *	0.74 (0.01)
USD 25,000–55,000	227 (32.2)	16.79 (0.24) ^‡^	23.32 (0.44) *	0.74 (0.01)
>USD 55,000	248 (32.2)	17.62 (0.24) ^‡^	24.25 (0.45) *	0.75 (0.00)
BMI, kg/m^2^				
<25	301 (34.4)	15.04 (0.18) ^‡^	20.04 (0.31) ^‡^	0.77 (0.01) ^‡^
25–30	218 (33.5)	17.22 (0.20) ^‡^	24.35 (0.38) ^‡^	0.72 (0.01) ^‡^
≥30	207 (32.1)	19.35 (0.23) ^‡^	27.10 (0.46) ^‡^	0.73 (0.01) ^‡^
Body fat percentage (%)				
<25	229 (31.4)	18.52 (0.21) ^‡^	27.79 (0.38) ^‡^	0.67 (0.00) ^‡^
25–35	230 (31.9)	16.94 (0.26) ^‡^	23.24 (0.42) ^‡^	0.74 (0.00) ^‡^
≥35	267 (36.7)	15.54 (0.19) ^‡^	19.62 (0.29) ^‡^	0.80 (0.01) ^‡^
Smoking				
Nonexposed	109 (21.0)	15.91 (0.35) ^‡^	21.45 (0.60) ^‡^	0.76 (0.01)
Expose to ETS	356 (50.6)	16.74 (0.19) ^‡^	23.23 (0.35) ^‡^	0.74 (0.00)
Active smokers	257 (28.4)	17.63 (0.21) ^‡^	24.39 (0.39) ^‡^	0.74 (0.01)
Alcohol consumption (drinks/year)				
<12	124 (29.5)	16.34 (0.36) ^‡^	b21.87(0.62) ^‡^	0.77 (0.01) ^‡^
≥12	429 (70.5)	17.43 (0.17) ^‡^	24.23 (0.31) ^‡^	0.74 (0.00) ^‡^

*, *p* < 0.05; ^‡^, *p* < 0.005 (tested by Student’s 2-tailed *t*-test or by one-way analysis of variance); Abbreviations: ETS, environmental tobacco smoke.

**Table 3 ijerph-19-06964-t003:** Linear regression coefficients (S.E.) of body measures with a one-unit increase in ln-DEHP metabolite concentrations in adults, with results weighted for sampling strategy.

Body Measures		MEHP (μg/g Creatinine)	*p*	MEOHP (μg/g Creatinine)	*p*	MEHHP (μg/g Creatinine)	*p*	MECPP (μg/g Creatinine)	*p*	Σ DEHP (μmol/g Creatinine)	*p*
Body weight (kg)											
Model 1	1563/193,092,305	0.092 (0.501)	0.856	2.020 (0.760)	0.018	2.047 (0.648)	0.007	2.003 (0.766)	0.019	2.031 (0.749)	0.016
Model 2	852/117,709,513	0.023 (0.452)	0.961	1.679 (0.572)	0.010	1.704 (0.538)	0.006	1.520 (0.616)	0.026	1.627 (0.593)	0.015
Body mass index (kg/m^2^)											
Model 1	1563/193,092,305	0.060 (0.157)	0.711	0.679 (0.211)	0.006	0.696 (0.190)	0.002	0.709 (0.227)	0.007	0.705 (0.217)	0.005
Model 2	852/117,709,513	0.052 (0.145)	0.726	0.598 (0.174)	0.004	0.617 (0.177)	0.003	0.578 (0.200)	0.011	0.602 (0.189)	0.006
Waist (cm)											
Model 1	1563/193,092,305	−0.012 (0.409)	0.978	1.549 (0.534)	0.011	1.532 (0.468)	0.005	1.512 (0.515)	0.010	1.533 (0.521)	0.010
Model 2	852/117,709,513	0.059 (0.414)	0.888	1.374 (0.423)	0.005	1.342 (0.403)	0.005	1.209 (0.391)	0.007	1.305 (0.411)	0.006
Subscapular Skinfold (mm)											
Model 1	1255/1,158,342,664	−0.362 (0.210)	0.104	0.082 (0.274)	0.770	0.151 (0.272)	0.586	0.133 (0.278)	0.640	0.073 (0.284)	0.801
Model 2	687/97,486,158	−0.423 (0.206)	0.058	0.019 (0.258)	0.943	0.113 (0.258)	0.669	0.038 (0.244)	0.877	0.005 (0.260)	0.985
Triceps Skinfold (mm)											
Model 1	1390/172,886,004	−0.253 (0.180)	0.181	0.263 (0.228)	0.267	0.237 (0.199)	0.254	0.261 (0.256)	0.323	0.237 (0.241)	0.342
Model 2	763/106,411,256	−0.198 (0.164)	0.245	0.272 (0.216)	0.227	0.234 (0.200)	0.262	0.205 (0.234)	0.395	0.216 (0.223)	0.347

Model 1 was adjusted for age, gender, race/ethnicity, educational level, household income, smoking status, and metabolic equivalent intensity level of activity. Model 2 was adjusted for Model 1 plus caffeine intake, total energy intake, total protein intake, total sugar intake, total carbohydrate intake, and total saturated fatty acid intake. Abbreviations: DEHP, di-(2-ethylhexyl) phthalate; MEHP, mono(2-ethylhexyl) phthalate; MEHHP, mono(2-ethyl-5-hydroxyhexyl) phthalate; MEOHP, mono(2-ethyl-5-oxohexyl) phthalate; MECPP, mono(2-ethyl-5-carboxypentyl) phthalate. Σ DEHP was the sum of (MEHP/278) + (MEHHP/294) + (MEOHP/292) + (MECPP/308) and corrected for urine creatinine.

**Table 4 ijerph-19-06964-t004:** Linear regression coefficients (S.E.) of bioelectrical impedance analysis parameters with a one-unit increase in ln-DEHP metabolite concentrations in adults, with results weighted for the sampling strategy.

Bioelectrical Impedance Analysis		MEHP (μg/g Creatinine)	*p*	MEOHP (μg/g Creatinine)	*p*	MEHHP (μg/g Creatinine)	*p*	MECPP (μg/g Creatinine)	*p*	Σ DEHP (μmol/g Creatinine)	*p*
Lean mass (kg)											
Model 1	674/96,499,431	−0.012 (0.320)	0.970	0.654 (0.444)	0.162	0.692 (0.382)	0.090	0.562 (0.481)	0.262	0.635 (0.444)	0.173
Model 2	415/61,088,209	−0.020 (0.296)	0.948	0.608 (0.384)	0.134	0.640 (0.364)	0.099	0.497 (0.416)	0.251	0.579 (0.393)	0.162
Fat mass (kg)											
Model 1	674/96,499,431	0.806 (0.610)	0.206	1.646 (0.675)	0.028	1.579 (0.617)	0.022	1.639 (0.697)	0.033	1.666 (0.695)	0.030
Model 2	415/61,088,209	1.000 (0.546)	0.087	1.716 (0.602)	0.012	1.560 (0.580)	0.017	1.547 (0.639)	0.029	1.651 (0.636)	0.020
Percent body fat (%)											
Model 1	674/96,499,431	0.419 (0.377)	0.285	0.800 (0.383)	0.034	0.718 (0.368)	0.040	0.808 (0.374)	0.047	0.794 (0.391)	0.036
Model 2	415/61,088,209	0.632 (0.352)	0.093	0.951 (0.337)	0.013	0.789 (0.336)	0.033	0.865 (0.348)	0.025	0.894 (0.354)	0.023
Cell membrane capacitance (nF)											
Model 1	674/96,499,431	−0.024 (0.028)	0.416	−0.008 (0.022)	0.716	−0.001 (0.018)	0.952	−0.014 (0.021)	0.525	−0.009 (0.020)	0.660
Model 2	415/61,088,209	−0.022 (0.029)	0.469	−0.002 (0.018)	0.925	0.004 (0.018)	0.824	−0.009 (0.020)	0.653	−0.004 (0.020)	0.843
ECF (L)											
Model 1	674/96,499,431	0.134 (0.101)	0.203	0.321 (0.149)	0.049	0.308 (0.133)	0.035	0.316 (0.167)	0.078	0.327 (0.156)	0.044
Model 2	415/61,088,209	0.147 (0.085)	0.105	0.289 (0.126)	0.037	0.271 (0.119)	0.038	0.279 (0.143)	0.069	0.292 (0.134)	0.045
ICF (L)											
Model 1	674/96,499,431	−0.106 (0.151)	0.493	0.196 (0.191)	0.321	0.230 (0.162)	0.175	0.139 (0.203)	0.505	0.179 (0.186)	0.351
Model 2	415/61,088,209	−0.120 (0.147)	0.430	0.189 (0.168)	0.278	0.224 (0.160)	0.184	0.123 (0.178)	0.498	0.168 (0.169)	0.337
ECF/ICF ratio											
Model 1	471/68,616,209	0.008 (0.003)	0.010	0.008 (0.002)	0.004	0.006 (0.002)	0.021	0.009 (0.003)	0.005	0.008 (0.003)	0.007
Model 2	415/61,088,209	0.009 (0.003)	0.003	0.007 (0.002)	0.004	0.005 (0.002)	0.041	0.009 (0.002)	0.002	0.008 (0.002)	0.004

Model 1 was adjusted for age, gender, race/ethnicity, educational level, household income, smoking status, and metabolic equivalent intensity level for activity. Model 2 was adjusted for Model 1 plus caffeine intake, total energy intake, total protein intake, total sugar intake, total carbohydrate intake, and total saturated fatty acid intake. Abbreviations: DEHP, di-(2-ethylhexyl) phthalate; MEHP, mono(2-ethylhexyl) phthalate; MEHHP, mono(2-ethyl-5-hydroxyhexyl) phthalate; MEOHP, mono(2-ethyl-5-oxohexyl) phthalate; MECPP, mono(2-ethyl-5-carboxypentyl) phthalate; ECF, extracellular fluid; ICF, intracellular fluid. Σ DEHP was the sum of (MEHP/278) + (MEHHP/294) + (MEOHP/292) + (MECPP/308) and corrected for urine creatinine.

**Table 5 ijerph-19-06964-t005:** Linear regression coefficients (S.E.) between ln-Σ DEHP and body composition parameters in different subpopulations of sample subjects with results weighted for sampling strategy.

	BMI (kg/m^2^)		Percent Body Fat (%)		ECF/ICF Ratio	
	βcoeff (S.E.)	*p*-Value	βcoeff (S.E.)	*p*-Value	βcoeff (S.E.)	*p*-Value
Age, year						
18–39	0.42 (0.37)	0.267	1.06 (0.38)	0.014	0.006 (0.003)	0.044
≥40	0.84 (0.31)	0.016	−0.03 (0.57)	0.960	0.010 (0.008)	0.228
Gender						
Men	0.48 (0.39)	0.246	−0.19 (0.52)	0.722	0.002 (0.006)	0.711
Women	0.77 (0.40)	0.073	1.78 (0.45)	0.001	0.011 (0.004)	0.011
Race						
Non-Hispanic White	0.39 (0.18)	0.048	0.98 (0.36)	0.016	0.012 (0.003)	0.001
Others	1.38 (0.46)	0.009	0.63 (0.72)	0.398	−0.001 (0.003)	0.658
Smoking						
Nonactive smokers	0.50 (0.20)	0.024	0.65 (0.36)	0.089	0.009 (0.002)	0.002
Active smokers	0.83 (0.48)	0.103	0.79 (0.48)	0.118	0.007 (0.006)	0.284

Adjusted for Model 2. Abbreviations: BMI, body mass index; DEHP, di-(2-ethylhexyl) phthalate; ECF, extracellular fluid; ICF, intracellular fluid.

## Data Availability

The datasets analyzed during the current study are available at the NHANES website (https://www.cdc.gov/nchs/nhanes/index.htm (accessed on 4 June 2022)).

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
