# Peer review of "Urine Di-(2-ethylhexyl) Phthalate Metabolites Are Independently Related to Body Fluid Status in Adults: Results from a U.S. Nationally Representative Survey"

_ijerph, 2022, doi:10.3390/ijerph19126964_

Round 1
Reviewer 1 Report
Thank you for inviting me to review the article titled “Urine di-(2-ethylhexyl) phthalate metabolites are independently related to body fluid status in adults: Results from a U.S. nationally representative survey”
I understand that the data used is around 20 years old now. However, that does not mean the authors have not done thorough work. I commend the authors for their hard work.
Criticisms:
Major: Study does not provide sufficient evidence to prove a correlation between ECF/ICF ratio and eDEHP
While the p-value may be significant, that does not mean the model has provided the appropriate correlation.
e.g., Correlation values near 1 are considered the best possible fit.
In table 5, if I am reading the table correctly, the regression coefficient values ( s.e.) are 0.003 with a p-value of 0.044; while this may be statistically significant, a fit of 0.003 reports that the model almost fails to fit the data.
This is true for all values for ECF/ICF ratio in table 5, which may also be due to the limited sample size, and thus this should not be reported as a positive correlation. The study has not generated sufficient evidence for this and this correlation should be removed, rather this correlation should be considered a hypothesis-generating finding. Yes, it should be mentioned if the available data was the limiting factor.
In contrast, body fat % for women and eDEHP have a relatively good correlation.
Minor:
The table labeling is a bit tough to understand.
Table 1: I could not understand the numbers, eg. The Title mentioned MEHP(ug/g), and the numbers were 2.13(1.03); if 1.03 is ug/g, what is the unit of measurement of MEHP. Please clarify better.
Same for tables 2, 3, and 4. Column headings need to be clearer.
Reviewer 2 Report
Major points:
- Abstract presents some results and conclusions, but they are all descriptive. What can we learn from these results and conclusions?
- For the introduction, could the authors highlight why the DEHP is selected and studied in a better way? Or please give some descriptions about the importance of DEHP in a clearer way.
- This paper is descriptive, could authors give a conclusion and biological meaning from your data at the end of the discussion?
Minor points:
1. Line 38, it is not clear why phthalates increase flexibility.
2. Line 39, is phthalate detected or are the metabolites of phthalate detected?
3. Table 1 and 2, suggest separating each category by a line
4. Table 2, why dose the lower BMI people have higher ratio? But higher body fat level shows higher ratio?
